# Particle Inertia Effects on Radar Doppler Spectra Simulation

Zeen Zhu[1], Pavlos Kollias[1,2] and Fan Yang[1]

[1]Environmental and Climate Sciences Dept, Brookhaven National Laboratory, Upton, NY, USA

[2] Division of Atmospheric Sciences, Stony Brook University, Stony Brook, NY, USA

*Correspondence to: Zeen Zhu (zzhu1@bnl.gov)*

**Abstract.** Radar Doppler spectra observations provide a wealth of information about cloud and precipitation microphysics and dynamics. The interpretation of these measurements depends on our ability to simulate these observations accurately forward. The effect of small-scale turbulence on the radar Doppler spectra shape has been traditionally treated by implementing the convolution process on the hydrometeor reflectivity spectrum and environment turbulence. This approach assumes that all the particles in the radar sampling volume respond the same to turbulent scale velocity fluctuations and neglects the particle inertial effect. Here, we investigate the inertia effects of liquid phase particles on the forward modelled radar Doppler spectra. A physics-based simulation is developed to demonstrate that big droplets, with large inertia, are unable to follow the rapid change of velocity field in a turbulent environment. These findings are incorporated to a new radar Doppler spectra simulator. Comparison between the traditional and the newly formulated radar Doppler spectra simulators indicates that the conventional simulator leads to an unrealistic broadening of the spectrum, especially in strong turbulence environment. This study provides clear evidence to illustrate the droplets inertial effect on radar Doppler spectrum and develops a physics-based simulator framework to accurately emulate the Doppler spectrum for a given Droplet Size Distribution in turbulence field. The proposed simulator has various potential applications for the cloud/precipitation studies and provides a valuable tool to decode the cloud microphysical and dynamical properties from Doppler radar observation.

## 1 Introduction

The radar Doppler spectrum represents the frequency (velocity) distribution of the backscattered radar signal at a particular range. For a vertically pointing radar, the Doppler spectrum provides the distribution of the backscattered signal over a range of Doppler velocities, whose value depends on the dynamical (i.e., vertical air motion) and cloud microphysical (i.e., hydrometeors concentration and sizes) properties within the radar sampling volume. A variety of research applications that utilize the full radar Doppler spectrum have been developed. For instance, Doppler spectrum can be used to retrieve rain Droplet Size Distribution (DSD) (Atlas et al., 1973), remove clutters and identify hydrometeor signals (Williams et al., 2018;Luke et al., 2008;Moisseev and Chandrasekar, 2009), identify drizzle development stage (Zhu et al., 2022;Acquistapace et al., 2019), retrieve vertical air motion (Kollias et al., 2002;Williams, 2012;Zhu et al., 2021), characterize the melting-layer properties (Li and Moisseev, 2020;Mróz et al., 2021), and to improve the representation of cloud microphysical process in model (Kollias et al., 2011b). Combined with the depolarization capability, Doppler spectrum can also be used for cloud-phase classifications and to investigate ice-cloud microphysical process (Luke et al., 2010;Luke et al., 2021;Kalesse et al., 2016;Oue et al., 2018). The forward Doppler spectra simulator can further be utilized to simulate radar observation from the modeling output to evaluate the model performance (Oue et al., 2020;Mech et al., 2020;Silber et al., 2022). The list of widely application of the Doppler spectrum in the cloud-precipitation research mentioned above is by no means exhaustive.

Despite the extensive applications, an unambiguous interpretation of radar Doppler spectrum still remains a challenging task in the cloud radar community. One important reason is a lack of full understanding of the entanglement between the hydrometeor microphysics and environment dynamics as well as their manifestation on the Doppler spectrum morphology (Kollias et al., 2002). More specifically, the Doppler spectrum width is mainly contributed by the spread of the still-air hydrometeor terminal velocity, the horizontal and vertical wind shear within the radar observation volume and the environment turbulence; while the Doppler frequency shift is a combined measure of the air motion and the particles falling velocity (Doviak, 2006). A successful separation of the microphysical and dynamical contributions to Doppler spectrum is essential to reduce retrieval uncertainties and to better characterize the cloud-precipitation properties (Zhu et al., 2021).

Doppler spectrum simulators have been invaluable for the interpretation of the radar
Doppler spectrum shape (Capsoni et al., 2001;Oue et al., 2020;Kollias et al., 2011a;Maahn et al.,
2015). Traditionally, the impact of turbulence on the shape of the radar Doppler spectrum is
represented by the convolution of the still air (no air motion) hydrometeor reflectivity spectrum
with a Gaussian distribution (Gossard and Strauch, 1989). The width of the Gaussian distribution
is parameterized as a function of the radar parameters and the turbulence intensity often
represented in terms of eddy dissipation rate (Borque et al., 2016). This approach is only valid
under the assumption that the droplet inertia effect is negligible and droplets with different sizes
can follow exactly the environment wind field. In reality, however, big droplets with large inertia
cannot follow the rapid change of wind velocity field unlike small droplets perform (Yanovsky,
1996;Lhermitte, 2002). Not accounting for the particle inertia effect can lead to a misinterpretation
of the Doppler spectrum and cause large uncertainties for retrieval products (Nijhuis et al., 2016).
Several physics-based frameworks have been proposed to simulate the droplet motions in
turbulence field (Khvorostyanov and Curry, 2005;Lhermitte, 2002). Here, the approach proposed
by Lhermitte (2002) is used to illustrate the droplets inertial effect and to investigate this effect on
the radar Doppler spectrum. In detail, we aim to answer the following questions: 1) How does
inertia affect the response of a droplet in a fluctuating turbulent wind field? 2) Is this effect
significant on the simulated and observed radar Doppler spectrum? and 3) How can we account
for the droplet inertia in radar Doppler spectrum simulators? Building on these investigations, a
new approach to generate radar Doppler spectrum is described.
The structure of this paper is organized as follows: section 2 describes the physical
modeling framework used to simulate the liquid droplet motion and to illustrate the droplets inertia
effect in a turbulent environment; section 3 proposes the physics-based Doppler spectrum
simulator and compares the emulated spectra to the ones generated from the traditional method;
in section 4 one observed Doppler spectrum is used as an illustrative example to compare the
Doppler spectrum generated from the two simulators; section 5 concludes the major results of this
study and followed by a discussion.




**2 Droplets inertial effect in a turbulent environment**
In this section, a physics-based simulation framework used to illustrate the droplets inertia
effect in a turbulent environment is presented. First, we will introduce the equations used to
describe the velocity of droplets moving in the air. Then a generated turbulent wind field is applied
to the simulation framework to illustrate the droplet inertial effect and the potential implication on
the generated Doppler spectrum.

**2.1 Motion of droplets in the air**
The fundemental dynamical framework of describing the droplets motion in the air is
adapted from Lhermitte (2002), p.81. Assuming a liquid droplet with diameter of $D$ , the motion
of the droplet in the air can be described as:

$$F - mg = m\frac{dV_D}{dt}$$

( 1)

where $m$ is the droplet mass, $V_D$ is the droplet velocity, $F$ is the drag force exerted by wind
expressed as:

$$F = \frac{C_d S(V_w - V_D)^2 \rho_a}{2} \cdot \text{sgn}(V_w - V_D)$$

( 2)

Where $C_d$ is the wind drag coefficient, $\rho_a$ is air density, $S$ is the droplet cross section normal to
wind direction. $V_w$ is wind velocity and $(V_w - V_D)$ indicates droplet velocity with respective to air.
In a turbulent environment, $V_w$ cloud be either positive or negative, thus the exerted wind can either
accelerate or decelerate the droplet velocity. To this end, the sign function $\text{sgn}(V_w - V_D)$ is
included to account for the wind drag force direction.
For spherical droplets, $S$ can be calculated as:

$$S = \frac{\pi D^2}{4}$$

( 3)


and droplet mass ($m$) is calculated as:

$$m = \frac{1}{6}\pi \rho_l D^3$$

( 4)

where $\rho_l$ is liquid water density.

The only unknown factor is the drag coefficient $C_d$, which should be derived from

experiment. Numerous studies have been conducted to measure the sphere terminal velocity in
fluid and estimate $C_d$ as a function of Reynolds number ($Re$) (Schlichting and Kestin, 1961;Lapple
and Shepherd, 1940;Haider and Levenspiel, 1989). However, the derived $C_d$ - $Re$ relationships in
the previous studies are applied for rigid spherical particles. For the rain droplets with large
diameter, the droplet is distorted and the exerted drag coefficient for a given $Re$ deviates from the
rigid sphere. To this end,  the drag term of the rain droplet is obtained from the measurement of
the terminal velocity of liquid droplets. Here,  we adapt the experiment data from Gunn and Kinzer
(1949), in which study $C_d$ and $Re$ are estimated for liquid droplets with diameter ranging from 100
$\mu m$ to 5.8 $mm$. The experiment-derived $C_d$ and $Re$ are shown in Figure 1, we further fit the data
with a fifth-degree polynomial (red line) to estimate $C_d$ for a given $Re$:

$logC_d = 1.4277 - 0.8598 \times logRe + 0.0699 \times (logRe)^2 - 0.0023 \times (logRe)^3 -$       ( 5)

$0.0003 \times (logRe)^4 + 0.0013 \times (logRe)^5$

Where the Reynolds number  $Re$ is represented as:

$Re = \dfrac{|V_w - V_D| D \rho_a}{\mu}$           ( 6)


where  $\mu$ is the air dynamic viscosity. The values used for $\rho_a$, $\rho_l$, and $\mu$ are $1.22\ kg\ m^{-3}$,
$1000\ kg\ m^{-3}$, $1.81 \times 10^{-5}\ kg\ m^{-1}\ s^{-1}$ , corresponding to atmospheric environment of $15°C$ and
$1000\ hPa$.
Combining (1)-(6), a set of ordinary differential equation is constructed, the droplet velocity ($V_D$)
for a given droplet with diameter $D$ as a function of time can be  resolved numerically for a given
wind field ($V_w$).





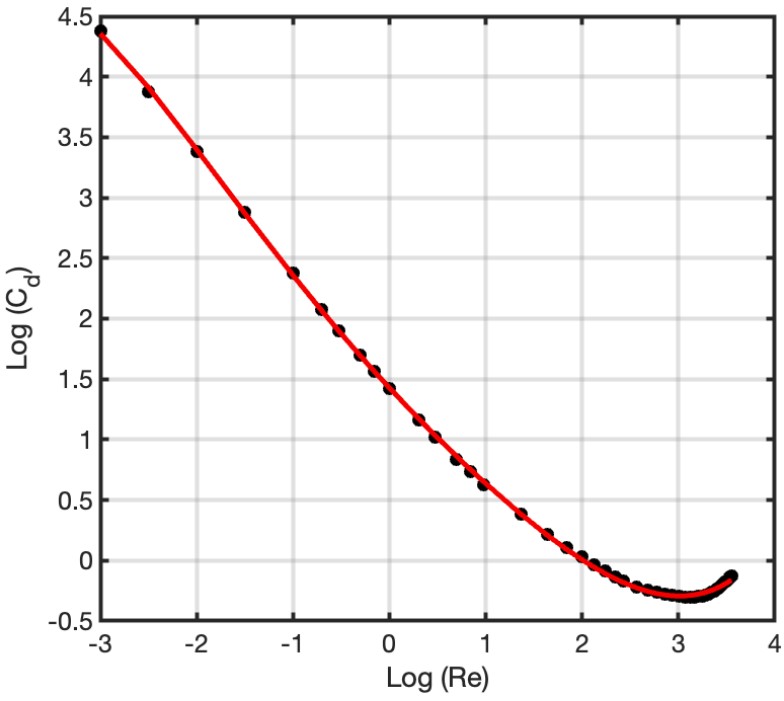


Figure 1: The black dots represent the experiment-derived $C_d$ and $Re$ adapted from Gunn and
Kinzer (1949). The red line is a fifth-degree polynomial fitting function.

## 2.2 Illustration of droplet inertial effect

We first illustrate the inertial effect by calculating droplets motion using a constant wind
velocity. For simplicity, here we assume all the droplets are moving horizontally, thus the gravity
($mg$) is neglected in Eq.1. Seven droplets with diameters of 10 $\mu m$, 50 $\mu m$, 100 $\mu m$, 500 $\mu m$, 1
$mm$, 2 $mm$, 5 $mm$ are selected to cover the size range of cloud droplet, drizzle and raindrops.
Initial velocity of all the droplets is 0 ms$^{-1}$, a constant wind velocity with 10 ms$^{-1}$ is exerted upon
the droplets when t > 0 s. Due to the wind drag force, droplets start to move but with different
accelerations depending on droplet inertia: droplets with small inertia are accelerated more quickly
than larger ones. This effect is clearly illustrated in Figure 2: droplet with diameter of 10 $\mu m$
quickly reach to the wind velocity within only 0.002s, while droplets with 1 $mm$ and 5 $mm$ need
5 and 50s to adjust their motion to the exerted wind velocity. The different response time of
droplets with different sizes to the exerted wind velocity suggests that small droplets are more
capable to follow the velocity variation than their large counterparts.

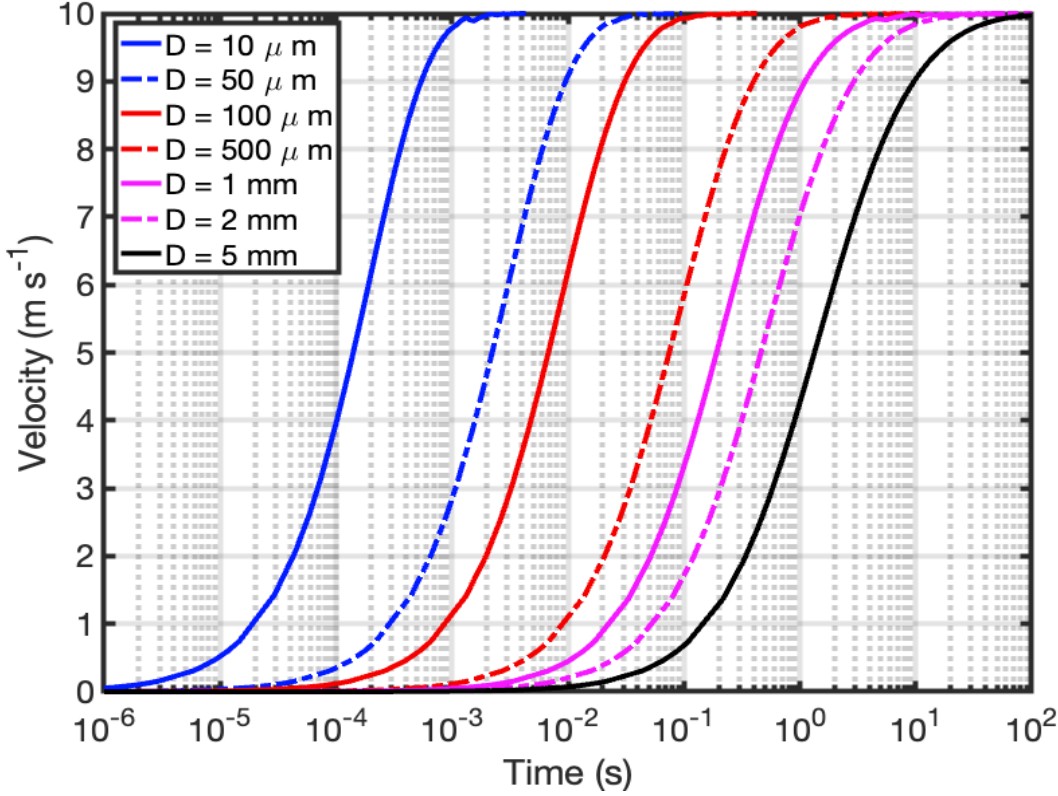


Figure 2. Velocity of droplets with diameter of 10 $\mu m$ (blue solid line), 50 $\mu m$ (blue dash-dot line),
100 $\mu m$ (red line), 500 $\mu m$ (red dash-dot line), 1 $mm$ (magenta solid line), 2 $mm$ (magenta dash-
dot line) and 5 $mm$ (black solid line) as function of time after exerted by a constant wind with 10
ms$^{-1}$ velocity.

170   In real atmosphere, air velocity is not constant but fluctuates with time as a representative

of turbulent nature. In this study we adapt the approach proposed by Deodatis (1996) by using the
Spectral Representation Method (SRM) to generate the turbulent wind field based on a predefined
Von Karman energy spectrum. The SRM is widely used in the wind engineering community due
to its high accuracy, simplicity and computational efficiency. (Shinozuka and Deodatis, 1991;Zhao
et al., 2021). Here, the 1-D turbulence wind is generated with 2 Hz sampling frequency, 1000s
duration and with standard deviation of 0.3 ms$^{-1}$, the codes being applied to generate the wind can
be accessed from Cheynet (2020). The selection of 0.3 ms$^{-1}$ standard deviation is based on a
quantitatively estimation of cloud radar observation under a typical cloudy environment.
Specifically, for the convective cloud system with eddy dissipation rate ($\varepsilon$) of $5 \times 10^{-3}$ m$^2$ s$^{-3}$
(Mages et al., 2022), the turbulence-contributed  Doppler spectrum width ($\sigma_t$) from a vertical
pointing radar with 30m range resolution($\Delta R$) and 0.3° beamwidth ($\theta$) at 1km height is estimated
to be 0.27 ms[-1] based on the equation from Borque et al. (2016):

$$\varepsilon \approx \frac{\sigma_t^3}{\sigma_z(1.35\alpha)^{3/2}}\left(\frac{11}{15} + \frac{4}{15}z^2\frac{\sigma_x^2}{\sigma_z^2}\right)^{-3/2}$$
(7)

Where $\alpha$ is the Kolmogorov constant with 0.5, $\sigma_z = 0.35 * \Delta R$, $\sigma_x = \dfrac{\theta}{4\sqrt{ln2}}$ , $\theta$ is the one-way
half-power width with unit of radian. $z$ is height above surface.

The spectrum and time series of the generated air velocity are shown in Figure 3: the

turbulence spectrum (Figure 3a) characterizes typical inertial subrange of the turbulence scale with
a standard deviation of 0.3 ms[-1](Figure 3b).

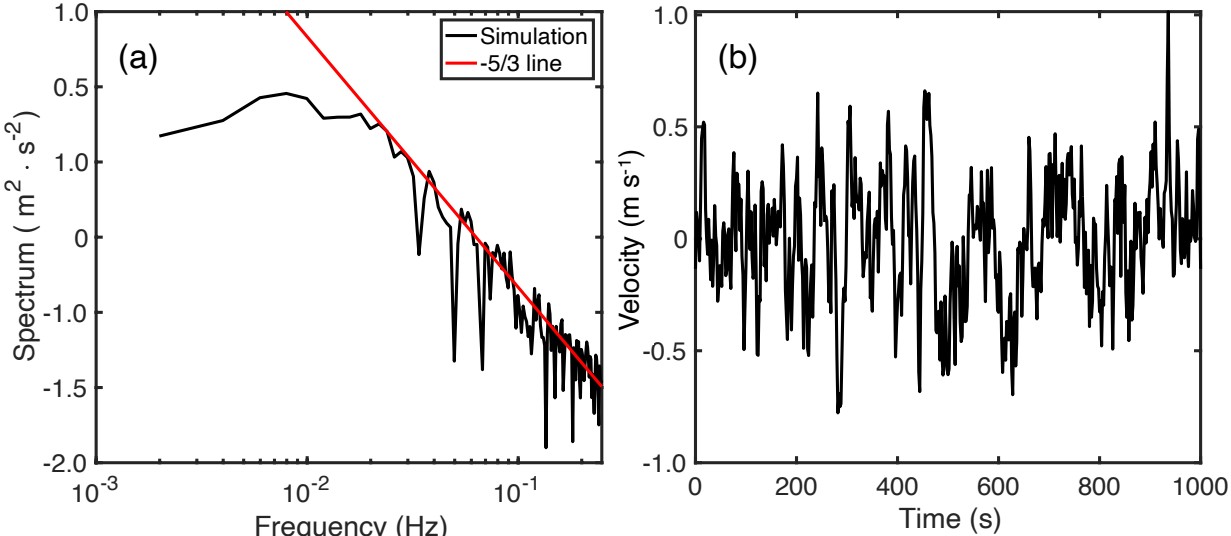


Figure 3. (a) Spectrum of the simulated turbulence (black line), red line represents the -5/3 slope.
(b): Time series of vertical velocity for the simulated turbulence.

The generated air velocity is assigned to $V_w$ in Eq. (2) to simulate the motion of droplets

with initial velocity set as 0 ms[-1]. Figure 4a shows the time-depended velocity of droplets with
selected diameter of 0.5 $mm$, 1 $mm$, 2 $mm$, 3 $mm$. Droplets with different sizes response
differently with the change of wind velocity, and there are two notable characteristics due to the
inertial effect (highlighted in the black oval in Fig. 4a). First, large droplets need longer time to
adjust to the wind velocity, thus there is a distinct time-lag when the peak velocity is reached for
different particles. Second, in addition to the time-lag, the peak velocity reached by the large
droplets is smaller than the small droplets. Here, we use correlation coefficient between the actual
wind velocity and the droplet velocity to quantify the inertial effect. A correlation coefficient of 1
represents droplets can follow exactly the wind velocity and a correlation coefficient less than 1
indicates a time-lag effect between the wind and droplet velocity due to droplet inertia. Figure 4b
shows that the correlation coefficient is close to 1 when the droplets are smaller than 50 μm but it
decreases dramatically as droplet size increases. The correlation coefficient reaches to 0 when
diameter reaches to 2000 $\mu m$. In addition, for droplets with diameters smaller than 300 $\mu m$ the
standard deviation of the actual droplet velocity is 0.29 ms$^{-1}$ (blue curve, Figure 4b), which is close
to standard deviation of the background wind field (0.3 ms$^{-1}$). As droplet size increases, the
velocity variation decreases due to droplet inertial effect.
The simulation results shown in Figure 4 suggest that droplets with diameter smaller than
300 $\mu m$ are less affected by inertia and can quickly adjust their velocity to the imposing wind field,
and thus, small cloud droplets can be treated as perfect air tracers (Kollias et al., 2001). On the
other hand, large droplets (D > 0.5 mm) exhibit a time lag in their response to the air motion and
an amplitude reduction (inertia-based filtering). As the observed Doppler velocity is a combined
measure of the droplet velocity and the ambient air motion, this droplet inertial effect is expected
to have a considerable effect on the generated radar Doppler spectrum. In the following section,
we will illustrate how the radar Doppler spectrum is affected by droplet inertia and how to account
for this effect using a new radar Doppler spectrum simulator.

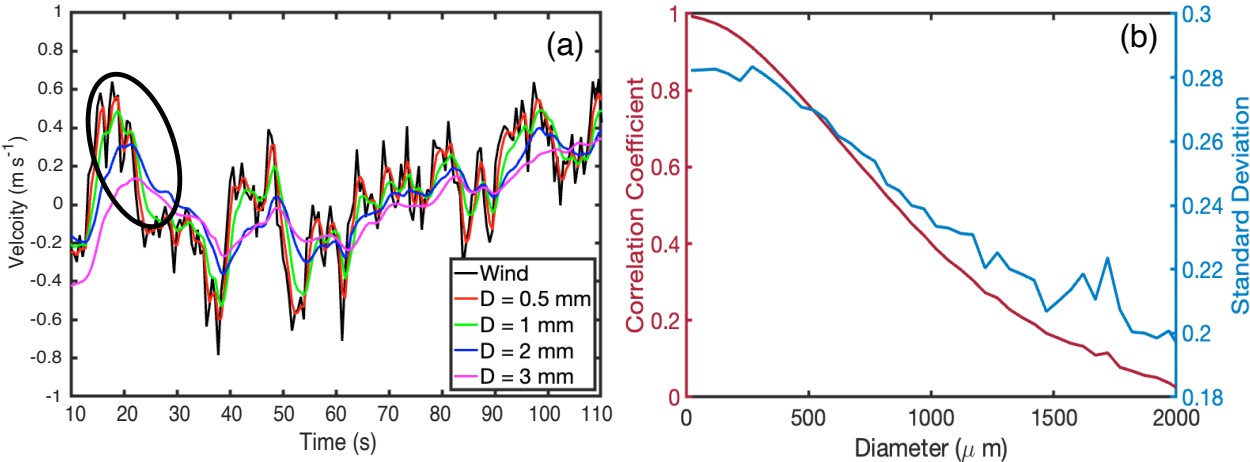




Figure 4. (a) Generated wind velocity field (black line) and the simulated velocity for particles with diameter of $0.5mm$ (red line), $1mm$ (green line), $2mm$ (blue line) and $3mm$ (magenta line) from 10s to 110s. The black oval indicates the period showing droplet inertia effect. (b) Left axis: correlation coefficient between wind field and droplet velocity for different droplets size; right axis: standard deviation of the droplets velocity with different droplets size. Only droplets with size from $0\ \mu m$ to $2000\ \mu m$ are shown for the sake of clarity.

## 3 Radar Doppler spectrum Simulator

Two methodologies for simulating the radar Doppler spectrum for a given DSD and turbulence conditions are used here. The first approach is the traditional one. All droplets, independent of their sizes, are assumed to have no inertial effects and thus act like perfect tracers. In this case, the radar Doppler spectrum in a turbulent environment is represented through the convolution of a Gaussian distribution and the radar Doppler spectrum in still air which is only determined by the hydrometeor DSD (Gossard, 1981; Kollias et al., 2011, Zhu et al., 2021). A brief overview of the traditional method is described in section 3.1.

### 3.1 Traditional Doppler spectrum simulator

For a given DSD described by a number concentration $N(D)$ per unit of volume in m$^{-4}$, the radar reflectivity $d\eta(D)$ (m$^2$/m$^3$) from particles with diameter between $D$ to $D + dD$ can be expressed as Lhermitte, (2002, p. 228):

$$d\eta(D) = N(D)\sigma_b(D)dD \qquad (8)$$

where $\sigma_b(D)$ is the backscatter cross section (m$^2$) of a particle with diameter D in m. Mie scattering theory is used to estimate $\sigma_b(D)$. In this formulation, the radar power spectrum distribution is provided in terms of particle size. Profiling radar do not observe the radar backscatter-energy power spectrum $d\eta(D)$ but the radar Doppler spectrum density $S_q(V_t)$ where $V_t$ is the droplet still-air terminal velocity. The conversion from droplet size to velocity requires a $V_t(D)$ relationship. Here, the function proposed by Lhermitte, (2002, p.120) is used to estimate $V_t$ as a function of droplet diameter ($D$):

$$V_t(D) = 920 \times (1 - exp(-6.8 \times D^2 - 4.88 \times D)) \qquad (9)$$

where the unit of $D$ and $V_t$ is in cm and cms$^{-1}$ respectively. Subsequently, the radar Doppler spectral density $S_q(V_t)$ in units of m$^2$m$^{-3}$ (ms$^{-1}$)$^{-1}$ is given by:

$$S_q(V_t) = \frac{d\eta}{dV_t} = \frac{d\eta}{dD}\frac{dD}{dV_t} = N(D)\sigma_b(D)\frac{dD}{dV_t} \qquad (10)$$
where $\frac{dD}{dV_t}$ is estimated from Eq. 9.
The $S_q(V_t)$ is the "still-air" radar Doppler spectrum where the only velocity contribution
is the droplet still-air terminal velocity . In the real atmosphere, the observed velocities from the
radar include the turbulent motions with scales larger or smaller than that of the radar sampling
volume (Kollias et al., 2001;Borque et al., 2016). The contribution of turbulence on Doppler
spectrum broadening is commonly parameterized as $\sigma_t$. It is important to note that the $\sigma_t$ value
also strongly depend on the radar sampling characteristics (Kollias et al., 2005). For the same EDR
value, $\sigma_t$ is lower for radar systems with short time dwell, narrow beamwidth and short pulse
length (Borque et al., 2016). The $\sigma_t$ is typically used to introduce the effect of turbulence on the
radar Doppler spectrum. Under the assumption of isotropic turbulence, the distribution of the
turbulent motions within the radar sampling volume can be approximated using a Gaussian
function:
$$G(v) = \frac{1}{\sigma_t\sqrt{2\pi}} \times \exp\left(-\frac{1}{2}\left(\frac{v}{\sigma_t}\right)^2\right) \qquad (11)$$
And its impact on radar Doppler spectrum is formulated by the convolution between $S_q(V_t)$ and
*G(v)* (Gossard and Strauch, 1989) as:
$$S(v) = (S_q * G)(v) = \int_{-\infty}^{\infty} S_q(u)G(v-u)du \qquad (12)$$
**3.2 Physics-simulation based Doppler spectrum simulator**
In this approach, instead of using a Gaussian distribution to parameterize turbulence field
and applying the convolution process to represent the interaction between DSD and environmental
turbulence, the radar Doppler spectrum is generated using a large number of simulated droplet
velocities during a given simulation period. Specifically, for droplet with diameter of *D* moving in
a turbulent flow, the droplet velocity at each specific time can be numerically resolved as $V(D,t)$
based on the ordinary differential equations described in section 2.1.
The radar Doppler spectrum density at each time step $S_t(v)$ can be directly estimated as:
$$S_t(v) = \frac{\sum N(D_{V_{i-1}\sim V_i})\sigma_b(D_{V_{i-1}\sim V_i})}{V_i - V_{i-1}} \qquad (13)$$
Where $D_{V_{i-1}\sim V_i}$ represents the diameter of the particle with velocity within the predetermined
Doppler velocity interval $[V_{i-1}, V_i]$ at each timestep, $N(D_{V_{i-1}\sim V_i})$ and $\sigma_b(D_{V_{i-1}\sim V_i})$ indicate the
number concentration and the backscatter power corresponding to each diamater. The
predetermined Doppler velocity $V_i$ is depended on the radar configuration of Nyquist velocity
($V_{nyquist}$) and the number of the Fast Fourier Transform points ($NFFT$):
$$V_i = -V_{nyquist} + \frac{2V_{nyquist}}{NFFT} \times i \; ; \; i = [1, 2, 3, \dots NFFT]$$ ( 14)
The final Doppler spectrum can be obtained by averaging $S_t(v)$ during the simulated period:
$$S(v) = \frac{1}{N_t} \sum_{t=1}^{t=N_t} S_t(v)$$ (15)
where $N_t$ is the total simulation timesteps:
$$N_t = T \times f$$ ( 16)
Where $T$ and $f$ is the simulated time and the sampling frequency of the generated turbulence
wind field.

It is noted that the emulated radar Doppler spectrum is dependent on the generated

turbulence flow, which is contolled by three parameters: time duration ($T$), sampling frequency ($f$)
and standard deviation ($\sigma$). $\sigma$ quantify the turbulence intensity while $T$ and $f$ determine the total
emulated time steps. Here we use the typical cloud radar configurations to guide the choice of $T$
and $f$. Specifically, $T$ is set as 2s and $f$ is set as 20 Hz to accommodate the cloud radar operated
at Atmospheric radiation measurement (ARM) program with approximately 40 spectra being
averaged in 2s (Kollias et al., 2005).

**3.3 Doppler spectra comparison from two simulators**

Both simulators described above are applied to emulate the Doppler spectrum observed by

a 94-GHz (W-band) profiling cloud radar for a given DSD and for a set of different turbulence
environments. The Nyquist velocity is set as $\pm$ 12 ms$^{-1}$ and a 512-point Fast Fourier Transform
(FFT) is used to generate the radar Doppler spectrum. The Marshall-Palmer exponential DSD
(Marshall and Palmer, 1948) with $N(D) = N_0 e^{-\Lambda D}$ is used to represent the DSD in the radar
sampling volume. The values of the intercept parameter $N_0$ and the slope factor $\Lambda$ are chosen to be
0.08 $cm^{-4}$ and 15 $cm^{-1}$. Droplet diameter ranges 10 to 4000 $\mu m$ with bin size as 1 $\mu m$. The
selection of W-band radar and the use of a rain DSD is because it is well known that the W-band
radar Doppler spectrum in rain has distinct features which allow to pinpoint the Doppler spectrum
morphology. Specifically, according to the Mie scattering theory, radar backscattering cross

section varies in an oscillatory manner with particle size (Mie, 1908). With the 3.2 mm wavelength radar, the backscattering cross section as a function of droplet size is characterized as several local minimal values with diameter of 1.66, 2.86 mm, which are corresponding to still-air terminal fall velocity of 5.83, 7.89 ms$^{-1}$. This unique feather is known as "Mie notches" in the radar Doppler spectrum (Kollias et al., 2002;Kollias et al., 2007;Courtier et al., 2022). In the simulation, turbulence field is generated with 20 Hz frequency ($f$ ),100s duration ($T$) and standard deviation ($\sigma$) of 0.05 ms$^{-1}$, 0.25 ms$^{-1}$, 0.35 ms$^{-1}$and 0.45 ms$^{-1}$, respectively.The reason of applying different turbulence settings is to better illustrate the droplet inertia effect under different turbulence environment. It is expected that with increasing turbulence intensity the droplet inertia effect will be manifested in larger differences between the generated radar Doppler spectrum from two methods.

When solving the ordinary differential equations described in Section 2.1, the initial droplet velocity is set as 0 ms$^{-1}$, thus at the beginning of the simulation the droplet gravity force is greater than the wind drag force, the droplet will accelerate until their terminal fall velocity is reached, after which the droplets fluctuate around the terminal fall velocity with variations induced by the exerted wind. The radar Doppler spectrum should be estimated after the steady state is reached. Here, we split the 100s simulated period to two parts: the first 40s is the "speed-up" time which allows the droplets of different size adjust to their steady state, the remaining 60s is used for Doppler spectrum emulation. Specifically,each Doppler spectrum is estimated within a 2s interval as illustrated in Section 3.2, then the generated 30 Doppler spectra in the 60s are further averaged to produce the final Doppler spectrum. This final average step is used to smooth the Doppler spectrum generated in a short period (2s) during which the averaged exerted wind may have a non-zero value.

The emulated Doppler spectrum from two methods with four turbulence settings are shown in Figure 5. In a turbulent environment with $\sigma_t$ of 0.05 ms$^{-1}$ (Figure 5a), the two simulated spectra (red and blue line in Figure 5a) and the Doppler spectrum without turbulence broadening (black line) are almost overlapping with each other, indicating that the radar Doppler spectrum shape is dominated by the DSD shape and the droplets still-air terminal fall velocity in weak turbulence conditions. For $\sigma_t$ equal to 0.25 ms$^{-1}$, the broadening of the right edge of the radar Doppler spectrum from the physics-based simulation(PBS) approach (red line in Figure 5b) is less than that produced with the convolution approach (blue line in Figure 5b). As $\sigma_t$ increases to 0.35 ms$^{-1}$, a

large differences between the right edges of the spectra from the two simulators can be clearly identified. When $\sigma_t$ reaches to 0.45 ms$^{-1}$, the right edge velocity difference between two spectra is larger than 1 ms$^{-1}$. Overall, the right edge from the PBS-generated Doppler spectrum is more steep than that from the covolution-based approach, illustrating that large droplets can not follow the rapidly changed turbulent field due to the inertia effect. Another notable finding is the left part of Doppler spectra (velocity smaller than 4 ms$^{-1}$) from two simulators almost overlap with each other in different turbulence scenarios, as this part of the spectrum is mostly contributed by small droplets with negligible inertial effect, thus the corresponding Doppler spectrum can be adequately represented by the convolution process.

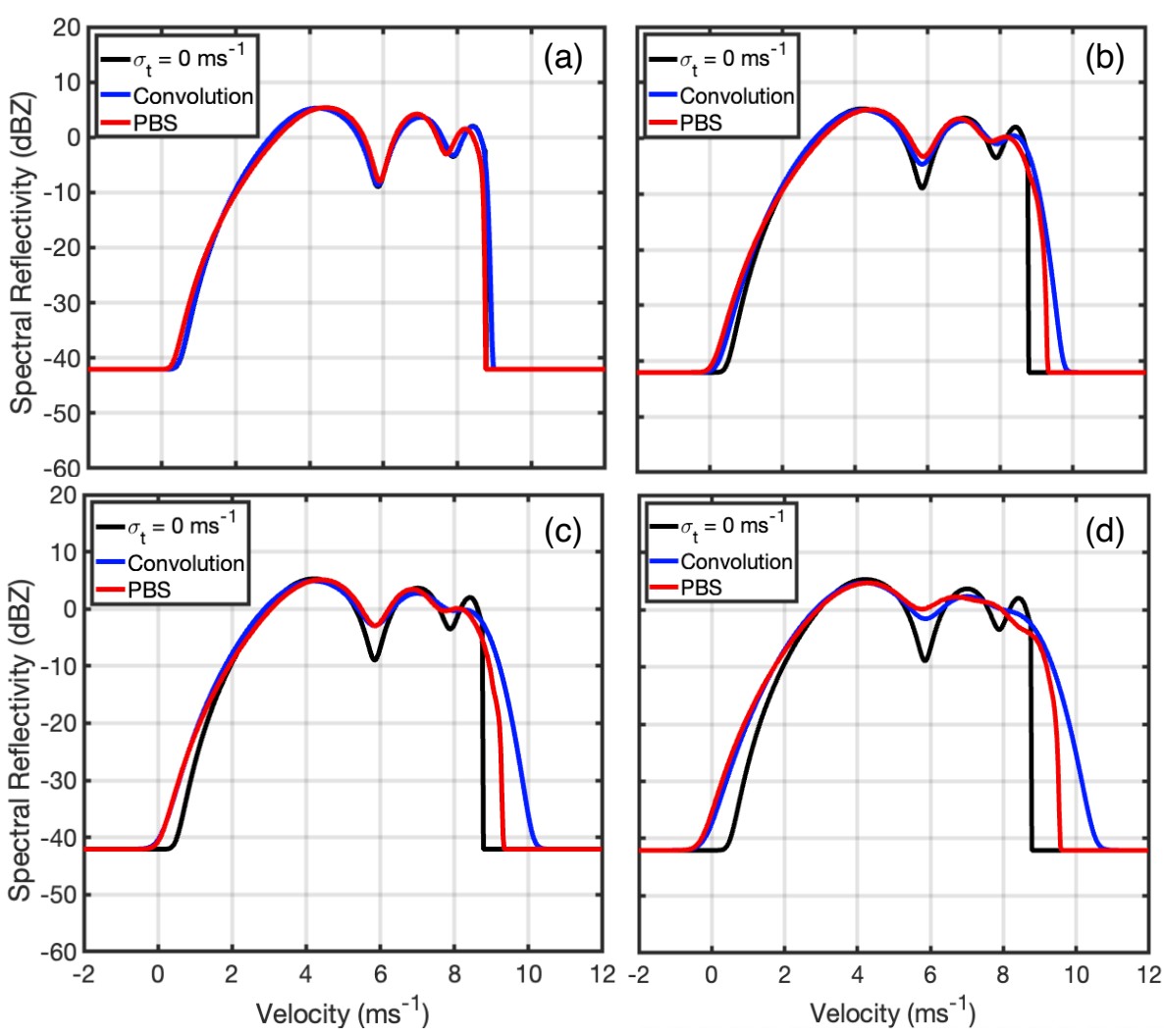

Figure 5. Doppler spectrum generated by the convolution-based (blue line) and physics-based simulation (PBS) (red line) approach for turbulence standard deviation with (a) 0.05 ms$^{-1}$, (b) 0.25

ms$^{-1}$, (c) 0.35 ms$^{-1}$, (d) 0.45 ms$^{-1}$. The black line represents generated Doppler spectrum with $\sigma_t =$
0 ms$^{-1}$. Positive velocity indicates downward motion.

Comparing the three generated Doppler spectra in Figure 5, we can clearly identify the
effect of droplet inertia on Doppler spectrum morphology under different turbulence environments.
In general, both simulators indicate a wider Doppler spectrum under a large turbulence condition,
but with different broadening magnitudes. The convolution-based approach generates a wider
spectra in a more turbulent environment. This overestimation of the turbulence broadening effect
indicates that the convolution process used in the conventional simulator is unable to accurately
represent the interaction between DSD and turbulence field. On the other hand, for the small
droplets, the inertial effect is negligible and the generated Doppler spectra from two approaches
are consistent with each other.  It is therefore concluded that the convolution process can simulate
the Doppler spectrum for the light drizzle precipitation which mostly occurs in marine boundary
layer clouds but it is inadequate to emulate Doppler spectrum for the heavy precipitation in deep
convection, especially in the presence of strong turbulence environment.

**4 An illustrative example of Doppler spectrum comparison between observation and**
**simulation**
In this section, we will present an illustrative example by using one observed Doppler
spectrum to evaluate the performance of the simulators. The observed Doppler spectrum is
obtained from the W-band ARM Cloud Radar (WACR) at the ARM Southern Great Plain (SGP)
observatory during a heavy precipitation period on May 9, 2007. For the WACR, the maximum
unambiguous velocity is 7.8ms$^{-1}$, which is smaller than the still-air terminal velocity of droplets
with diameter larger than 3$mm$ and lead to velocity folding.  Here velocity de-aliasing process is
performed to reconstruct the Doppler spectrum with velocity from 0 ms$^{-1}$ to 11 ms$^{-1}$. The observed
Doppler spectrum is further calibrated from the displacement caused by vertical air motion by
pinpointing the location of first Mie notch of the Doppler spectrum to 5.83ms$^{-1}$.(Kollias et al.,
2002). To simulate the Doppler spectrum, the hydrometeor DSD and the turbulence broadening
term ($\sigma_t$) are needed. Here, the raindrops DSD is observed from the impact disdrometer which can
measure droplet diameter from 0.3$mm$ to 5.4 $mm$ with 20 bins (Wang et al., 2021). The temporal
resolution of the WACR and the disdrometer is 4.28s, 1min respectively. To make the observation
from two instruments comparable, the WACR-observed Doppler spectra are averaged over 1min
to coincide with the disdrometer observational period. For this example, we use the disdrometer-
measured DSD from 05:44 to 05:45 UTC to simulate the radar Doppler spectrum and compare it
with the one observed of WACR in the same period.
The observed DSD is shown in Figure 6a, and the corresponding WACR-observed Doppler
spectrum is shown as the black line in Figure 6b. Based on the observed DSD, the radar Doppler
spectrum for the droplets falling in still air is generated (not shown), from which the DSD-
contributed Doppler spectrum width ($\sigma_D$) is estimated as 1.34 ms$^{-1}$. Since the wind shear
broadening contribution ($\sigma_S$) to radar Doppler spectrum  is generally smaller than $\sigma_D$ and the
turbulence broadening ($\sigma_t$) (Borque et al., 2016), here we neglect the $\sigma_S$ contribution and  estimate
$\sigma_t$ as:
$$\sigma_t^2 = \sigma_O^2 - \sigma_D^2$$

Where $\sigma_O$ is the observed Doppler spectrum width, which is 1.46 ms$^{-1}$ in this example, and
$\sigma_t$ is estimated as 0.58 ms$^{-1}$. To estimate the accuracy of $\sigma_t$, we further assume the observed DSD
is the only source of the uncertainty. Considering that  the accuracy of the droplets size
measurement of the disdrometer is approximately $\pm5\%$ (Wang et al., 2021), the uncertainty of $\sigma_D$
and $\sigma_t$ is estimated as 0.15 ms$^{-1}$.
With the observed DSD and the estimated $\sigma_t$, the radar Doppler spectrum can be simulated.
It is noted that large rain droplets falling in the air are nonspherical, thus backscattered power from
an oblate droplet may be different from the one from rigid liquid sphere. To this end, for the Mie
scattering calculation, axis ratio ($\frac{a}{b}$) of the droplet with diameter largher than 2mm is considered
as a function of diameter (*D*) with unit of *mm* (Pruppacher and Beard, 1970):
$$\frac{a}{b} = 1.03 - 0.062D$$

The simulated Doppler spectrum from the  convolution and the PBS method are shown in
Figure 6(b). It is noticeable that the Doppler spectrum from the PBS approach (red line) is more
noisy than that from the convolution approach (blue line). This is due to the insufficient bin
categories of the particle measured from disdrometer, it is expected that with increasing the number
of measured particle size, the generated Doppler spectrum become more smooth. Nevertheless, it
is still recognizable that the both the morphology and the magnitude of the PBS-based spectrum
right edge is more consistent with observation compared with the one generated from the

convolution approach. Both of the two simulators represent the first peak of the Doppler spectrum from 3 ms$^{-1}$ to 6 ms$^{-1}$ very well, while neither of them generate a consistent second peak morphology compared with observation. The left edge of the Doppler spectrum from the convolution-based approach is broader than the observation, while the PBS is unable to represent the Doppler spectrum smaller than 1ms$^{-1}$ due to the abscent of the droplets with diameter smaller than 0.3 *mm* observed from disdrometer.

The purpose of this Doppler spectrum comparison is not for a robust validation but used as an illustrative example to show the morphology of the simulated Doppler spectrum based on real observations and to discuss the required measurements would be used for robust Doppler spectrum simulator validation. To a certain degree, a more consistency Doppler spectrum morphology is identified between the observation and from the PBS simulator, especially for the right edge of the spectrum. However, great cautions should be taken for further interpretation as both of the simulators cannot represent the left part of the Doppler spectrum and the second notches very well. This discrepancy is mainly because the observed DSD by disdrometer may not an adequate representation of the hydrometeors that contribute the Doppler spectrum observed by WACR. Specifically, there are three critical challenging issues should be overcome before a solid and convincing Doppler spectrum simulator evaluation effort being performed: 1) the disdrometer is located at the surface, while the lowest measurement height of WACR is 460m. When the rain droplets fall, droplets may collide, breakup, and being advected from adjacent regions by the horizontal wind; Thus a large uncertainty is expected by using the surface-observed DSD to represent the hydrometeor distribution at 450m above; 2) the observed DSD from the disdrometer only measure droplets with 20 size categories, which is insufficient for the physics-based simulation to generate a smooth and complete Doppler spectrum; 3) the uncertainty of the estimated $\sigma_t$ is challenging to be well constrained due to the large uncertainty of the observed DSD mentioned above. A comprehensive and solid validation of the Doppler spectrum simulator require simultaneous and well- aligned DSD and Doppler spectrum measurement; large number of the measured droplet size categories and carefully estimation of the environment turbulence broadening factors.

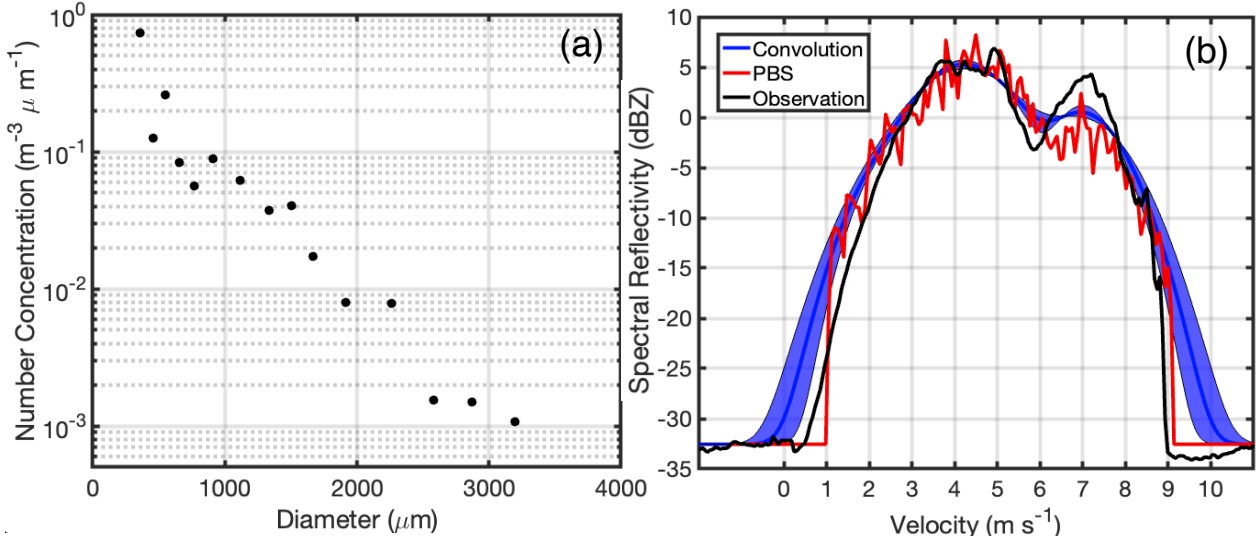

Figure 6. (a) Black dots represent the observed raindrop number concentration from disdrometer at 05:44 (UTC) on May 9, 2007 on SGP site. (b) Doppler spectra simulated from the PBS (red) and convolution (blue) method and the observed spectrum from WACR (black line). The blue shaded region represents the uncertainty of the simulated Doppler spectrum produced by the uncertainty in $\sigma_t$ based on the convolution method. Positive velocity indicates downward motion.

## 5 Conclusions

The radar Doppler spectrum offer unprecedent capabilities for studying cloud and precipitation microphysics. Recent advancements in radar technology and signal processing have enable the continuous recording of high-quality radar Doppler spectra observations from a wide range of profiling radar systems (Kollias et al., 2005;Kollias et al., 2016). Until now, the simulation of the radar Doppler spectra was based on well-established techniques (Gossard, 1988;Kollias et al., 2011a). However, inertial effect of large droplets are typically neglected in the design of current simulators. Here, the impact of the liquid droplet's inertia on the shape of the radar Doppler spectrum was investigated. A physics-based simulation framework is developed to simulated the droplets velocity in a given turbulence environment. It demonstrates that big droplets with large inertia will take longer time to adapt to the change of velocity field, indicating large droplets are incapable of following the turbulence wind as small droplets do.

Building on the simulation framework, a new approach is proposed to emulate Doppler spectrum by simulating the velocity of each droplet during the entire time domain. The simulated

W-band radar Doppler spectrum is compared with the one generated from the traditional method
for a typical DSD with four different turbulence environments.The comparison indicates that the
traditional Doppler simulator without considering the inertial effect generates an artificially
broader Doppler spectrum. This inertia effect becomes more noticeable as turbulence intensity
increases. This finding suggests that special caution should be taken when applying convolution-
based approaches to represent DSD-turbulence interaction in heavy precipitation. In the case of
light precipitation mostly happening in marine boundary layer cloud, the droplet inertia effect on
Doppler spectrum is negligible and the traditional simulator generates consistent results with the
proposed simulator.
One WACR-observed Doppler spectrum collected from the ARM SGP observatory is
compared with the simulated Doppler spectrum as an illustrative example to validate the fidelity
of the simulator from the convolution and the PBS-based approach. The presented case shows that
the proposed PBS generate a more similar morphology of the right edge of the Doppler spectrum
compared with the traditional simulator. However, both of two simulator fail to reconstruct the left
edge and the second notch of the Doppler spectrum. These inconsistents are due the fact that the
surface-based DSD from disdrometer is inadequate to represent the hydrometeor observed by
cloud radar at a high level. A careful and solid validation of the radar Doppler spectrum simulator
would require co-aligned observations of DSD and Doppler spectrum and well-constrained
turbulent broadening estimations. Nevertheless, the proposed Doppler spectrum simulator, with
the ability to simulate individual droplet motion as well as their manifestation on Doppler spectrum,
provide an valuable tool to improve the understanding of Doppler radar observation from a
fundemental physics perspective. We expect this proposed Doppler spectrum simulation
framework can stimulate more studies to better interpret the Doppler radar observation and to
decode the microphysics and dynamics information concealed in radar Doppler spectrum.

**Competing interests.**

**P. K.** is the associate editor of AMT and the peer-review process was handled by an independent
editor. The authors have no other competing interests to declare.


**Code/Data availability**

The codes of the proposed Doppler spectrum simulator can be accessed via

https://doi.org/10.5281/zenodo.7897981.

Ground-based data were obtained from the Atmospheric radiation measurement (ARM) user facility, a U.S. Department of Energy (DOE) Office of Science user facility managed by the Office of Biological and Environment Research.

W-Band (95 GHz) ARM Cloud Radar (WACRSPECCMASKCOPOL). 2007-05-09 to 2007-05-10, Southern Great Plains (SGP) Central Facility, Lamont, OK (C1). Compiled by K. Johnson, D. Nelson and A. Matthews. ARM Data Center. Data set accessed 2022-07-05 at http://dx.doi.org/10.5439/1025318.

Impact Disdrometer (DISDROMETER). 2007-05-09 to 2007-05-10, Southern Great Plains (SGP) Central Facility, Lamont, OK (C1). Compiled by D. Wang. ARM Data Center. Data set accessed 2022-07-05 at http://dx.doi.org/10.5439/1025181.

**Author contributions**

Zeen Zhu implemented the method, performed the analysis, produced the figures, and wrote the initial draft of the manuscript. Pavlos Kollias supervised and provided advice and guidance on all aspects of the analysis and contributed to the writing of the manuscript. Fan Yang advised on results interpretation and manuscript editing. All authors read the manuscript draft and contributed comments.

**Financial support**

Zeen Zhu's contribution is supported by Brookhaven National Laboratory via the Laboratory Directed Research and Development Grant LDRD 22-054. Pavlos Kollias and Fan Yang are supported by the US Department of Energy (DOE) under contract DE-SC0012704.

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
