# Peer review of "Particle Inertia Effects on Radar Doppler Spectra Simulation"

_Atmospheric Measurement Techniques, 2023_

## Referee Comment (RC3)

**Review**

This study examines the effect of particle inertia in forward modelling vertically pointing cloud radar Doppler spectrum. The authors have carried out theoretical analysis and validated their method with field observations. The logic of this manuscript is clear, and the research question is sound. Different approaches are clearly compared in good-quality figures. Relevant references have been cited.

However, there is one major flaw regarding the validation. Please see my comments below. In addition, the description of the new simulator is difficult to follow. Therefore, my recommendation is major revision.

**Major comments**

- Description of the new spectrum simulator is not clear to me. Since it is a new method, a detailed and explicit explanation is needed. I am wondering where is the term quantifying the turbulence. Also, Eq.12. looks the same as eq.9.
- 2. The validation part is questionable to me. The broadening effect seems to be exaggerated.

If I understood correctly, the authors assume no horizontal and shear winds. Then, eq 5 in Borque 2016 changes to  $\sigma^2 = \sigma_d^2 + \sigma_t^2$ .  $\sigma^2$  can be estimated from the observed spectrum, and the magnitude of  $\sigma_t^2$  depends on  $\sigma_d^2$ . If  $\sigma_d^2$  as retrieved from the surface DSD is underestimated,  $\sigma_t^2$  will be overestimated. Then, the broadening effect will be unrealistically large. In a word, the accuracy of  $\sigma_t^2$  depends on how well the raindrop spectrum was constructed from surface observations. As far as I could image, the fitting process may lead to the underestimation of  $\sigma_d^2$ . I believe the authors should carefully quantify the uncertainty of  $\sigma_d^2$  in the revised manuscript.

In addition, in Figure 5, what is the height of the observed spectrum? How well the DSD observed at surface can be used to simulate the spectrum aloft observed by a W band radar? In other applications, these two issues do not significantly contribute to retrieval errors. Given the change of DSD can significantly affect the evaluation results, I am afraid they should be well discussed in this study.

**Technical issues**

I have some suggestions for technical corrections, but I am not a native speaker.

- 1. Either using positive or negative to indicate downward is fine, but it is appreciated to make a statement in each figure's caption.
- 2. L22. consistent with
- 3. L27. applications for cloud/precipitation
- 4. L28. microphysical and dynamical
- 5. L34. For a vertical
- 6. L35. Provide
- L53. and many other places. Spectral broadening is contributed by a list of factors such as turbulence, horizontal wind, spectral window etc. In some cases, turbulence dominates this broadening effect.
- 8. L162. This work is published on a journal with which not many cloud radar people familiar, please detail this method.
- 9. L164-166. This sentence is confusing. Spectrum width is affected by hydrometeor size distribution,

how can it be a constant value?

- 10. L188. close.
- 11. Eq.13. where is n in  $S_t$ ?
- 12. L292. L306, and many other places. Turbulent environment
- 13. L306. echo?
- 14. L307. of
- 15. L338. spectra from two approaches are consistent with each other.
- 16. L340. add a comma before but
- 17. L343. approaches
- 18. L344. The black line
- 19. Figure4. dB(10log10 (mm6 m-3))
- 20. Figure4 and many other places. Simulated approach looks strange to me. I would call it physics-based approach.
- 21. L4225. Can be employed in more studies

---

## Author Comment (AC1)

This study examines the relevance of hydrometeor inertia for the accurate simulation of radar Doppler spectra. The authors develop a novel approach to simulate Doppler spectra based on the equations governing the motion of hydrometeors, and compare it with the traditionally used convolution-based approach. They find that the traditional approach tends to overestimate the degree of broadening associated with inertial particles, while it correctly represents the degree of broadening for particles whose inertia is low enough for them to act as tracers. The authors then compare spectra simulated both with their novel approach, and with the traditional approach, with one observed spectrum. The comparison displays a better matching between the observed spectrum and the spectrum simulated with the author's approach.

I find the research question of the study to be significant for the cloud radar community, and the novel approach developed by the authors to be sound. The figures are polished, and the overall structure of the manuscript is well layed-out, clearly displaying the reasoning process of the authors. However, I find several sentences in the manuscript to lack in clarity or to have been poorly written, and improvements in this regard are needed. See my detailed comments below.

The only major flaw that I found in the study is in the comparison with observations. The comparison was performed only for one observed sample (one Doppler spectrum). In my opinion such a limited validation is not sufficient to prove the accuracy of the approach developed by the authors. The validation with respect to observations needs to be considerably expanded, and should be performed on a statistical basis using a large number of Doppler spectra, if possible recorded during a few separate events.

I do understand that the authors might want to include such an extensive validation in a follow-up study, but if that is the case, the comparison with observations in section 5 should be framed as a simple illustrative exercise, instead of a proper validation analysis. I included below a list of changes that I deem necessary if the authors do not intend to expand the comparison with observations in the current study. I anyhow strongly recommend that such an extensive validation is included in the current manuscript, as it would make the whole study substantially more sound.

In conclusion, I recommend this study for publication in AMT after major revisions: clarity of the text needs to be improved and I recommend either that the comparison with observations is considerably expanded or the text is adjusted so that such comparison is not presented as a complete validation.

Response: We want to thank the reviewer for the detailed edits and for the constructive suggestions. We agree with reviewer that only one comparison example is insufficient to make a robust validation statement. A solid and comprehensive evaluation of the Doppler spectrum simulator would require observation from different cloud/precipitation scenarios: cloud-drops only, drizzle, light precipitation, heavy precipitation etc. However, as what we have discussed in the revised manuscript, such validation effort would require high-quality of DSD observation and turbulence broadening estimation. Unfortunately, the observation what we have obtained from the current instruments have relatively large uncertainties and cannot meet these high standards. To this end, we prefer to present this comparison section as an illustrative example, while with more focus being placed to the discussion of the uncertainties should be considered for a careful Doppler

spectrum simulator validation effort. We also hope this discussion can promote more suitable observational datasets in future field campaigns which can be used for robust Doppler spectrum simulator validation.

**Changes needed if the validation is not expanded**

The sentences at lines 19-23 in the abstract (from "Doppler spectra observed …" until "...morphology") need to be removed. The comparison with observations should not be mentioned in the abstract as it doesn't have scientific significance.

Response: The related sentences have been deleted from the abstract.

The sentence at lines 84-85 ("section 5 uses … simulator") needs to reflect the fact that the comparison presented in section 5 is not a validation, but a mere illustrative exercise. Additionally the phrasing "real observed Doppler spectra" is not accurate and should read "one real observed Doppler spectrum".

Response: Changes have been made in the revised manuscript:

Line 84: "…in section 4 one observed Doppler spectrum is used as an illustrative example to compare the Doppler spectrum generated from the two simulators…"

The text in section 2 should be moved to section 5 (or an appendix), as it is not relevant for the main topic of the manuscript, which is the approach development. The text could also be condensed.

Response: Section2 and section 5 in the previous manuscript are combined to section 4 in the revised manuscript.

The title as well as the text in section 5 should reflect the fact that only one observed spectrum is used in the comparison instead of multiple spectra. The singular "spectrum" should be used instead of the plural "spectra".

Response: The title of section 4 in the revised manuscript is modified as:

Line 368: "An illustrative example of Doppler spectrum comparison between observation and simulation"

The sentences at lines 373-377 need to be rephrased in a more careful manner, as the simple analysis shown does not provide enough evidence to support these statements.

The same applies to lines 410-415, 418-419, 421-422 in the conclusions.

The conclusions need to clearly state that the new approach needs to be systematically validated against observations, and that the applications suggested at lines 422-427 may only be looked into after the accuracy of the approach is demonstrated against observations.

Responses to the previous three comments:

We have incorporated the reviewer's comments and rephrased the Doppler spectrum comparison section (section 4) in the revised manuscript. We have clarified that the comparison is not used for validation purpose but serving as an illustrative example. We have also made a thorough discussion on the uncertainties involved in this validation framework and the datasets would be required for future validation effort.

Line 419: "…The purpose of this Doppler spectrum comparison is not for a robust validation but used as an illustrative example to show the morphology of the simulated Doppler spectrum in real environment and to discuss the required measurements needed for robust Doppler spectrum simulator validation. To a certain degree, a more consistency Doppler spectrum morphology is identified between the observation and from the PBS simulator, especially for the right edge of the spectrum. However, great cautions should be taken for further interpretation as both of the simulators cannot represent the left part of the Doppler spectrum and the second notches very well. This discrepancy is mainly because the observed DSD by disdrometer may not an adequate representation of the hydrometeors that contribute the Doppler spectrum observed by WACR. Specifically, there are three critical challenging issues should be overcome before a solid and convincing Doppler spectrum simulator evaluation effort being performed: 1) the disdrometer is located at the surface, while the lowest measurement height of WACR is 460m. When the rain droplets fall, droplets may collide, breakup, and being advected from adjacent regions by the horizontal wind; Thus a large uncertainty is expected by using the surface-observed DSD to represent the hydrometeor distribution at 450m above; 2) the observed DSD from the disdrometer only measure droplets with 20 size categories, which is insufficient for the physics-based simulation to generate a smooth and complete Doppler spectrum; 3) the uncertainty of the estimated $\sigma_t$ is challenging to be well constrained due to the large uncertainty of the observed DSD mentioned above. A comprehensive and solid validation of the Doppler spectrum simulator require simultaneous and well- aligned DSD and Doppler spectrum measurement; large number of the measured droplet size categories and carefully estimation of the environment…"

**Further scientific questions/issues**

Throughout the whole text the term "fall velocity" or "droplet velocity" is used as synonym for "still-air terminal velocity" (e.g. at lines 56, 195, 310, ...). This is incorrect and should be adjusted. Throughout the whole text the term "quiet air" is used (e.g. lines 63, 211, 243, ...). The term "still air" is far more commonly used and I recommend that this is used instead.

Response: We thank the reviewer's comments. We have adapted the term "still-air terminal velocity" where it applies in the manuscript.

Throughout the whole text the term "movement" is used as synonym for "motion" (e.g. at lines 82, 102, 105, ...). This is incorrect and should be adjusted.

Response: Corrections have been made throughout the manuscript.

Throughout the whole text the verb "resolve" is used when referring to simulated drop velocities (e.g. at lines 106, 215, 265, 270, 400, 418, ...). I find this ambiguous because the actual particle velocity is not observed in any way, but an artificial velocity value is produced and then used to calculate the Doppler spectrum. Therefore I suggest this is adjusted, e.g. by replacing "resolve" with "simulate".

Response: We thank the reviewer's suggestion. Corrections have been made in the manuscript where we consider it is appropriate for such replacement.

References should be given for all formulas in sections 3.1 and 4.1.

Response: References have been added to the equations.

Throughout sections 3.1 and 3.2 it is not stated in the text whether gravity was included in the simulations. I assume it is not since the corresponding term is missing in eq. (1). Its inclusion or omission should be stated explicitly. If it is included, eq. (1) should be adjusted accordingly.

Response: In the revised manuscript we have considered the gravity in the physical framework (Eq.1). The gravity is included in the Doppler spectrum simulation (section 3 and 4), but it is not included in section 2.2. We have clarified this statement in the revised manuscript:

Line 154: "…We first illustrate the inertial effect by calculating droplets motion using a constant wind velocity. For simplicity, here we assume all the droplets are moving horizontally, thus the gravity ($mg$) is neglected in Eq.1…"

Lines 160-162: since the calculation of the artificial wind time series is an integral part of the proposed approach, the method by Deodatis (1996) should be briefly summarized here.

Response: We have added more details of the wind generation process in the revised manuscript:

Line 172: "…In this study we adapt the approach proposed by Deodatis (1996) by using the Spectral Representation Method (SRM) to generate the turbulent wind field based on a predefined Von Karman energy spectrum. The SRM is widely used in the wind engineering community due to its high accuracy, simplicity and computational efficiency. (Shinozuka and Deodatis, 1991;Zhao et al., 2021). Here, the 1-D turbulence wind is generated with 2 Hz sampling frequency, 1000s duration and with standard deviation of 0.3 ms$^{-1}$, the codes being applied to generate the wind field can be accessed from Cheynet (2020)…"

The same applies to the method used for the calculation of the broadening term sigma_t by Borque et al. (2016), mentioned at lines 247-248.

Response: We have added the equation for $\sigma_t$ estimation in the revised manuscript:

Line 178: "…The selection of 0.3 ms$^{-1}$ standard deviation is based on a quantitatively estimation of cloud radar observation under a typical cloudy environment. Specifically, for the convective cloud system with eddy dissipation rate ($\varepsilon$) of $5 \times 10$-3 m$^2$ s$^{-3}$ (Mages et al., 2022), the turbulence-contributed Doppler spectrum width ($\sigma_t$) from a vertical pointing radar with 30m range resolution($\Delta R$) and 0.3° beamwidth ($\theta$) at 1km height is estimated to be 0.27 ms$^{-1}$ based on the equation from (Borque et al., 2016)

$$\varepsilon \approx \frac{\sigma_t{}^3}{\sigma_z(1.35\alpha)^{3/2}}\left(\frac{11}{15} + \frac{4}{15}z^2\frac{\sigma_x^2}{\sigma_z^2}\right)^{-3/2} \tag{1}$$

Where $\alpha$ is the Kolmogorov constant with 0.5, $\sigma_z = 0.35 * \Delta R$, $\sigma_x = \frac{\theta}{4\sqrt{ln2}}$, $\theta$ is the one-way half-power width with unit of radian. $z$ is height above surface…"

When citing a book (e.g. Lhermitte 2002) the exact chapter or pages should be indicated.

Response: Changes have been made in the revised manuscript.

Line 241: the radar reflectivity $d\eta(D)$ (m$^2$/m$^3$) from particles with diameter between $D$ to $D + dD$ can be expressed as (Lhermitte, 2002, p. 228):

Line 249: "…Here, the function proposed by (Lhermitte, 2002, p.120) is used to estimate $V_t$ as a function of droplet diameter ($D$)…"

Figure 3a: what values were assigned to the initial velocities of the droplets?

Response: Changes have been made in the revised manuscript.

Line 195: "…The generated air velocity is assigned to $V_w$(Eq. (2)) to simulate the motion of droplets with initial velocity set as 0 ms$^{-1}$…"

Eq. (12): the symbol S_t appears here for the first time and it should be introduced.

Response: We have rephrased the description of the simulator framework. More details can be found in Section 3.2 in the revised manuscript.

Eq. (12) and line 270: if I understand the text correctly, here $V_t$ is the turbulence-affected drop velocity, but the symbol $V_t$ was previously used to indicate the still-air terminal velocity. I believe a new or different symbol should be used here instead.

Response: We have rephrased the description of the simulator framework. More details can be found in Section 3.2 in the revised manuscript.

Line 271: please clarify what the term "DSD Doppler spectra" means.

Response: We have rephrased the description of the simulator framework. More details can be found in Section 3.2 in the revised manuscript.

Lines 294-297: this sentence should be split and expanded. First the concept of Mie notch should be introduced. Then the fact that the Mie notch can be used to compare the two approaches should be explained.

Response: We have modified this part in the revised manuscript:

Line 308: "…The selection of W-band radar and the use of a rain DSD is because it is well known that the W-band radar Doppler spectrum in rain has distinct feathers which allow to pinpoint the Doppler spectrum morphology. Specifically, due to the Non-Rayleigh scattering, the backscattered power for rain droplets with specific radius is identified as local minimal value, this characteristics is manifested as the "Mie notches" in the observed Doppler spectrum…"

Figures 4 and 5b: the labels to the y axes should indicate the name of the variable ("spectral reflectivity" in this case) in addition to the unit. The unit is also misindicated as "dB(10log(mm6 m-3))", it should read either "dBZ" or "10log(mm6 m-3)".

Response: Changes have been made in the revised manuscript.

Section 5: I would like more details on the processing of the observational radar data. Were the data corrected for attenuation? Was de-aliasing performed? Looking at Fig. 5b it seems that the spectrum was shifted to have its left edge at 0 m/s, is this the case? If the comparison with observations is expanded these details need to be included in the text, otherwise it is sufficient if they are only included in the authors' reply.

Response: The radar Doppler spectrum is not calibrated for attenuation. More details of the Doppler spectrum processing are added in the revised manuscript:

Line 373: "…For the WACR, the maximum unambiguous velocity is 7.8ms$^{-1}$, which is smaller than the still-air terminal velocity of droplets with diameter larger than 3$mm$ and lead to velocity folding. Here velocity de-aliasing process is performed to reconstruct the Doppler spectrum with velocity from 0 ms$^{-1}$ to 11 ms$^{-1}$. The location of the observed Doppler spectrum is further calibrated from the displacement caused by vertical air motion by pinpointing the location of first Mie notch of the Doppler spectrum to 5.83ms$^{-1}$ …"

Line 355 reports that the data were collected between 05:44 and 05:45, but line 94 reports a temporal resolution of 4.28 seconds. Were multiple spectra observed over that one minute averaged together? If not please indicate the exact timestamps with hours, minutes, and seconds. If yes, please clearly state it in the text.

Response: We have modified the corresponding part in the revised manuscript:

Line 381: "…The temporal resolution of the WACR and the disdrometer is 4.28s, 1min respectively. To make the observation from two instruments comparable, the WACR-observed Doppler spectra are averaged over 1min to coincide with the disdrometer observational period. For this example, we use the disdrometer-measured DSD from 05:44 to 05:45 UTC to simulate the radar Doppler spectrum and compare it with the one observed of WACR in the same period…"

**Stylistic/technical corrections**

We appreciate the reviewer's detailed edits. All the corrections have been made in the revised manuscript.

Line 40: I believe the correct phrasing is "...remove clutter and identify hydrometeor signal".
Line 44: I find the phrase "improve the microphysical medlin process" to be unclear.
Lines 49-50: I find the grammar in this sentence to be overall incorrect, and it should be rephrased.
Lines 53-55: I believe that the phrase "… spectrum is contributed by …" is gramatically incorrect and it should be improved.
Line 58: I believe the correct phrasing is "… to reduce retrieval uncertainties…".
Line 60: I believe the correct phrasing is "Doppler spectrum simulators".
Line 61: I believe the correct phrasing is "Doppler spectrum shape".
Line 69: "… unlike small doplets".
Line 71: "… large uncertainties for retrieval products".
Line 75: "How does inertia…".
Line 97: "… identify hydrometeor signals".
Line 97: "Additionally, an impact disdrometer…".
Line 98: "The disdrometer".
Line 136: "The values used for … are …".

Line 137: I find the phrase "...as a representation of environment…" unclear. I recommend that it is rephrased.

Line 145: "...cloud droplet, drizzle, …".

Line 160: "...a turbulent environment...".

Lines 160-161: I find the phrasing "...are equivalently inertia-free…" unclear.

Line 190: How small? Quantify please.

Line 202: "...wind field...".

Lines 210-212: This sentence is hard to follow and should be rewritten.

Line 221: "...is only applicable to vertical...".

Line 232: "...Doppler spectrum density … Vt is the droplet…".

Line 256: "...and its impact on radar Doppler…".

Line 274: I believe this line should read "… total number of simulated timesteps...".

Line 276: "...where T and f are ...".

Lines 287-288: I would rephrase "the values of the intercept parameter N0 and the slope factor Gamma are chosen to be ...".

Line 288: "… droplet diameter ranges…".

Line 293: "...larger differences between the generated…".

Lines 298-300: I find the term "adjusted time" unclear.

Lines 313-314: this sentence reads wrong and should be adjusted. E.g.: "… a large differences between the right edges of the spectra from the two simulators can be clearly identified."

Line 330: "Comparing the three ...".

Lines 353 and 371: Marshall is spelled with two l.

Line 367: "...spectral power compared to ...".

Line 370: I believe the sentence should read "… both the simulated Doppler spectrum and the convolution-based Doppler spectrum near the second notch are not consistent ...".

Lines 375-376: "… has shown significant improvement in correctly emulating…".

Line 387: either "Radar Doppler spectra ..." or "The radar Doppler Spectrum...".

Line 392: I would rephrase "...inertial effects are typically neglected…".

Lines 397-398: "… velocity field… incapable of following … as small droplets do.".

Line 406: "… caution should be taken when applying convolution-based approaches to represent ...".

Line 419 "… various potential ...".

Line 450: I believe this should read "initial draft".

Line 456: It should read either "contribution is" or "contributions are".

**Reference**

Borque, P., Luke, E., and Kollias, P.: On the unified estimation of turbulence eddy dissipation rate using Doppler cloud radars and lidars, Journal of Geophysical Research: Atmospheres, 121, 5972-5989, 2016.

Cheynet, E.: Wind field simulation (text-based input), Zenodo, Tech. Rep., 2020, doi: 10.5281/ZENODO. 3774136, 2020.

Deodatis, G.: Simulation of ergodic multivariate stochastic processes, Journal of engineering mechanics, 122, 778-787, 1996.

Mages, Z., Kollias, P., Zhu, Z., and Luke, E. P.: Surface-based observations of cold-air outbreak clouds during the COMBLE field campaign, Atmospheric Chemistry and Physics Discussions, 1-39, 2022.

Shinozuka, M., and Deodatis, G.: Simulation of stochastic processes by spectral representation, 1991.

Zhao, N., Huang, G., Kareem, A., Li, Y., and Peng, L.: Simulation of ergodic multivariate stochastic processes: An enhanced spectral representation method, Mechanical Systems and Signal Processing, 161, 107949, 2021.

---

## Author Comment (AC2)

Dear Dr. Davide Ori,

We appreciate the time and effort you have dedicated to providing detailed feedback on our manuscript. Your comments are valuable to improve the physical framework of our Doppler spectrum simulator. The detailed responses are shown below.

Response to the dynamical system comments:

In the original manuscript we adapted a simple and idealized assumption to simulate the droplets movement in the air: we assumed that all the droplets are moving horizontally, and the only force being exerted is the horizontal wind. To simulate the Doppler spectrum observed from a vertical pointing radar, we further assume that the droplets falling with their terminal velocity (V) is equivalent to exerting an additional horizontal wind (with speed as V) to the particle. In the revised manuscript, we adapt the reviewer's suggestions and include the particle gravity in Eq1. for the Doppler spectrum simulation. We also included a sign function to accounting for the wind force exerted from different direction detailed in Eq.2. The equation describing the motion of particle in a fluid is now consistent with previous study, e.g., Equation 3 in Businger (1965) and Equation 9.1 in Lamb and Verlinde (2011). A detailed description of the modified model can be seen in section 2.1 in the revised manuscript. The results shown in Figure 5 and Figure 6 are reproduced using the modified framework.

Response to the inconsistent terminal fall velocity comments:

The relationship between the drage coefficient ($C_d$) and Reynolds number ($R_e$) in the original manuscript is based on Schlichting and Kestin (1961), in which the relationship is fitted from experiment results where a rigid sphere falls in the fluid. This relationship is not applicable to non-spherical or distorted particles such as for raindrops with diameter larger than 2 $mm$. The terminal fall velocity used in the original manuscript (Eq.8) is one of the fitting function based on Gunn and Kinzer (1949) in which a carefully experiment is conducted to measure terminal fall velocity of liquid droplets terminal in atmosphere. In the reviewer's comments, we can notice a consistent terminal fall velocity between the experimental-based (i.e., Gunn and Kinzer, 1949) and the theoretical-derived (i.e., Schlichting and Kestin, 1961) method until rain drops are larger than 2 mm. To mitigate this discrepancy for larger rain drop, we utilized a new fitting function in the revised manuscript to describe the relationship between $C_d$ and $R_e$ based on the same experiment data to derive the terminal fall velocity(Gunn and Kinzer, 1949). This newly fitted $C_d$ - $R_e$ function can generate a consistent terminal fall velocity compared with the experimental results (Figure R1). We have modified the manuscript accordingly.

Line122: "…The only unknown factor is the drag coefficient $C_d$, which should be derived from experiment. Numerous studies have been conducted to measure the sphere terminal fall velocity in fluid and estimate $C_d$ as a function of Reynolds number ($Re$) (Schlichting and Kestin, 1961;Lapple and Shepherd, 1940;Haider and Levenspiel, 1989). However, the derived $C_d$ - $Re$ relationships in the previous studies are applied for rigid spherical particles. For the rain droplets with large diameter, the droplet is distorted and the exerted drag coefficient for a given $Re$ deviates

from the rigid sphere. To this end, the drag term of the rain droplet is obtained from the measurement of the terminal velocity of liquid droplets. Here, we adapt the experiment data from Gunn and Kinzer (1949), in which study $C_d$ and $Re$ are estimated for liquid droplets with diameter ranging from $100\ \mu m$ to $5.8\ mm$. The experiment-derived $C_d$ and $Re$ are shown in Fig. 1, we further fit the data with a fifth-degree polynomial (red line) to estimate $C_d$ for a given $Re$:…"

$$logC_d = 1.4277 - 0.8598 \times logRe + 0.0699 \times (logRe)^2 - 0.0023 \times (logRe)^3 - \qquad (1)$$
$$0.0003 \times (logRe)^4 + 0.0013 \times (logRe)^5$$

[Figure]

Figure R1: Droplet terminal fall velocity as a function of diameter from the experiment fitting (Lhermitte 2002) and from the theoretical estimation of the terminal fall speed. We adapted the reviewer's python code to generate Figure1, except a newly fitted $C_d$ - $R_e$ function is utilized. This newly fitted $C_d$ - $R_e$ function can generate a consistent terminal fall velocity compared with the experimental results.

**Minor Points:**

Title and Line 13 - the generic term" particle" suggest that the model is applicable to any hydrometeor. However, it seems clear to me that the proposed methodology is applicable only to liquid drops. Perhaps it is better to specifically address only liquid precipitation.

Response: We have clarified that the proposed simulator is applicable to liquid phase particles in the revised text.

Line 13: "…Here, we investigate the inertia effects of liquid phase particles on the forward modelled radar Doppler spectra…"

Line 454: "… Here, the impact of the liquid droplet's inertia on the shape of the radar Doppler spectrum was investigated…"

Line 53-56: I believe that there are some additional contributors to the spectral broadening. For example, the finite beamwidth allows for some of the horizontal wind component as well as the vertical shear of the horizontal wind to cause some spectral broadening.

Response: We have rephrased the sentence as follows:

Line 53:"…the Doppler spectrum width is mainly contributed by the spread of the hydrometers terminal velocity, the horizontal and vertical wind shear within the radar observation volume, and small-scale turbulence.."

Line 89 - The data section seems a little misplaced here, it makes a sudden interruption to the introductory argument which focuses on the methodology and the methodology itself which is presented in Sec 3. Sec. 2 is very short and the data are used only in section 5 which is again quite short. Since the method is the central focus of the paper I suggest to make Section 2 a subsection of the current Section 5.

Response: We have merged the previous data section and the Doppler spectrum comparison section in the revised manuscript as the reviewer suggested.

Line 102 -" turbulence" - turbulent

Response: Changes have been made in the revised manuscript.

Line 109 - The title of this subsection explicitly mention turbulence. However there is no effect of turbulence explicitly taken into account. The subsection merely list the equations used to define the dynamics of spherical objects in a fluid regardless of its laminar or turbulent status.

Response: The subsection title has been modified as:

Line 102: "Motion of droplets in the air"

Figure 3 - y-label velcoity - velocity

Response: Changes have been made in the revised manuscript.

Sec 4.2 (and partially also Fig 1) it is not clear to me how the equation of motion is resolved. Is a numerical method for the solution of ordinary differential equations used? What is the time resolution of the method? Is the power spectrum of turbulent air motion truncated at a certain frequency? what is the expected uncertainty in the determination of the drop speed?

Response: The ordinary differential equations described in section 2.1 are solved numerically, in this project we applied the Matlab function *ode45*. For the Doppler spectrum simulation, the utilized time resolution is 0.05s which is consistent with the frequency of the generated velocity field (20 Hz). The full spectrum of the generated turbulent air velocity is applied with no truncation in frequency. We have rephrased the description of the Doppler spectrum simulator in Section 3.2 in the revised manuscript.

Line 370 - I am not sure how the DSD shape might shift the location of the scattering notch. To me the notch occurs at a specific size and provided that there is a well-defined velocity-size relation it would occur at a specific velocity regardless of the DSD. DSD discrepancies might only move the notch up or down in the spectral power. At lines 229-230 it is stated that Mie scattering theory is used for the scattering computation which would imply perfectly spherical raindrops, However, I think that such big raindrops are not spherical but rather slightly oblate. This means that their length along the vertical (which is the one relevant for the Mie resonances considering the vertical propagation direction) is smaller. Thus, a larger oblate raindrop is needed to produce a Mie resonance effect along the vertical direction than a spherical one. I suggest the authors to try using a spheroidal approximation of raindrops for scattering.

Response: We thank the reviewer's suggestions. We have considered the oblate shape of the droplets for Mie scattering in the Doppler spectrum comparison section (Section 4) in the revised manuscript.

Line 400: "…With the observed DSD and the estimated $\sigma_t$, the radar Doppler spectrum can be simulated. It is noted that large rain droplets falling in the air are nonspherical, backscattered power from an oblate droplet may be different from the one from rigid liquid sphere. To this end, for the Mie scattering calculation, axis ratio ($\frac{a}{b}$) of the droplet with diameter larger than 2mm is considered as a function of diameter ($D$) with unit of *mm* (Pruppacher and Beard, 1970):…"

$$\frac{a}{b} = 1.03 - 0.062D$$

Code/Data availability - the authors include reference to a github repository owned by a person which is not listed among the co-authors. It is fine but I would suggest to include not the github repository, which is subject to modifications, but a more permanent link. Luckily, the repository offers also a packaged version that got a DOI on zenodo. It is ok to keep the reference in the data availability section, but zenodo offers the option to properly give author attribution, have it in the list of references, and to pin the citation to a permanent link of a specific version of the software.

I take the opportunity to also invite the authors to publish their code openly which would be of great benefit for the radar community and for the repeatability of their results. The AMT journal

invites all authors to publish their data and codes, and in this particular case it would have greatly helped in the understanding of what has been done in the study

Response: We thank the reviewer's suggestions. We have cited the codes used in the manuscript in the way provided by the author. The cited reference is linked to a zenodo page.

Line 177: "…the codes being applied to generate the wind can be accessed from Cheynet (2020)…"

We would like to publish our radar Doppler spectrum simulator codes once the revised manuscript addresses all the reviewer's concerns and no more changes will be made to the simulator.

**Reference**

Businger, Joost Alois. "Eddy diffusion and settling speed in blown snow." Journal of Geophysical research 70, no. 14 (1965): 3307-3313.

Cheynet, E.: Wind field simulation (text-based input), Zenodo, Tech. Rep., 2020, doi: 10.5281/ZENODO. 3774136, 2020.

Gunn, R., and Kinzer, G. D.: The terminal velocity of fall for water droplets in stagnant air, Journal of Atmospheric Sciences, 6, 243-248, 1949.

Haider, A., and Levenspiel, O.: Drag coefficient and terminal velocity of spherical and nonspherical particles, Powder technology, 58, 63-70, 1989.

Lapple, C., and Shepherd, C.: Calculation of particle trajectories, Industrial & Engineering Chemistry, 32, 605-617, 1940.

Pruppacher, H. R., and Beard, K.: A wind tunnel investigation of the internal circulation and shape of water drops falling at terminal velocity in air, Quarterly Journal of the Royal Meteorological Society, 96, 247-256, 1970.

Schlichting, H., and Kestin, J.: Boundary layer theory, Springer, 1961.

---

## Author Comment (AC3)

This study examines the effect of particle inertia in forward modelling vertically pointing cloud radar Doppler spectrum. The authors have carried out theoretical analysis and validated their method with field observations. The logic of this manuscript is clear, and the research question is sound. Different approaches are clearly compared in good-quality figures. Relevant references have been cited.

However, there is one major flaw regarding the validation. Please see my comments below. In addition, the description of the new simulator is difficult to follow. Therefore, my recommendation is major revision.

Response: We want to thank the reviewer's comments and suggestions. We have modified the validation and the methodology section in the revised manuscript. The detailed responses can be seen below.

**Major comments**
Description of the new spectrum simulator is not clear to me. Since it is a new method, a detailed and explicit explanation is needed. I am wondering where the term is quantifying the turbulence. Also, Eq.12. looks the same as eq.9.

Response: We have rephrased the description of the simulator in the revised manuscript, please refer to section 3.2 for more details.

The validation part is questionable to me. The broadening effect seems to be exaggerated. If I understood correctly, the authors assume no horizontal and shear winds. Then, eq 5 in Borque 2016 changes to $\sigma2 = \sigma d2 + \sigma t2$. $\sigma2$ can be estimated from the observed spectrum, and the magnitude of $\sigma t2$ depends on $\sigma d2$. If $\sigma d2$ as retrieved from the surface DSD is underestimated, $\sigma t2$ will be overestimated. Then, the broadening effect will be unrealistically large. In a word, the accuracy of $\sigma t2$ depends on how well the raindrop spectrum was constructed from surface observations. As far as I could image, the fitting process may lead to the underestimation of $\sigma d2$. I believe the authors should carefully quantify the uncertainty of $\sigma d2$ in the revised manuscript.

In addition, in Figure 5, what is the height of the observed spectrum? How well the DSD observed at surface can be used to simulate the spectrum aloft observed by a W band radar? In other applications, these two issues do not significantly contribute to retrieval errors. Given the change of DSD can significantly affect the evaluation results, I am afraid they should be well discussed in this study.

Response: We want to thank the review's comments. In the revised manuscript we directly utilized the observed DSD from the disdrometer to simulate the Doppler spectrum instead of using the Marshall-Palmer fitting function as in the previous manuscript. This change is intended to eliminate the DSD error caused by the fitting process. In the revised manuscript we highlight that the Doppler spectrum comparison shown in section4 is not used for validation purpose but as an illustrative example. We made a thorough discussion on the representative of the surface-observed

DSD for the W-band radar observation and the uncertainty of the $\sigma_t$ estimation in the revised manuscript:

Line 387: "…The observed DSD is shown in Figure 6a, and the corresponding WACR-observed Doppler spectrum is shown as the black line in Figure 6b. Based on the observed DSD, the radar Doppler spectrum for the droplets falling in still air is generated (not shown), from which the DSD-contributed Doppler spectrum width ($\sigma_D$) is estimated as 1.34 ms$^{-1}$. Since the wind shear broadening contribution ($\sigma_S$) to radar Doppler spectrum is generally smaller than $\sigma_D$ and the turbulence broadening ($\sigma_t$) (Borque, Luke et al. 2016), here we neglect the $\sigma_S$ contribution and estimate $\sigma_t$ as:

$$\sigma_t^2 = \sigma_O^2 - \sigma_D^2$$

Where $\sigma_O$ is the observed Doppler spectrum width, which is 1.46 ms$^{-1}$ in this example, and $\sigma_t$ is estimated as 0.58 ms$^{-1}$. To estimate the accuracy of $\sigma_t$, we further assume that the observed DSD is the only source of the uncertainty. Considering that the accuracy of the droplets size measurement of the disdrometer is approximately ±5% (Wang, Bartholomew et al. 2021), the uncertainty of $\sigma_D$ and $\sigma_t$ is estimated as 0.15 ms$^{-1}$
…"

Line 419: "…The purpose of the Doppler spectrum comparison is not for a robust validation but used as an illustrative example to show the morphology of the simulated Doppler spectrum in real environment and to discuss the required measurements would be used for robust Doppler spectrum simulator validation. To a certain degree, a more consistency Doppler spectrum morphology is identified between the observation and from the PBS simulator, especially for the right edge of the spectrum. However, great cautions should be taken for further interpretation as both of the simulators cannot represent the left part of the Doppler spectrum and the second notches very well. This discrepancy is mainly because the observed DSD by disdrometer may not an adequate representation of the hydrometeors that contribute the Doppler spectrum observed by WACR. Specifically, there are three critical challenging issues should be overcome before a solid and convincing Doppler spectrum simulator evaluation effort being performed: 1) the disdrometer is located at the surface, while the lowest measurement height of WACR is 460m. When the rain droplets fall, droplets may collide, breakup, and being advected from adjacent region by the horizontal wind; Thus a large uncertainty is expected to use the surface-observed DSD to represent the hydrometeor distribution at 450m above; 2) the observed DSD from the disdrometer only measure droplets with 20 size categories, which is insufficient for the physics-based simulation to generate a smooth and complete Doppler spectrum; 3) the uncertainty of the estimated $\sigma_t$ is challenging to be well constrained due to the large uncertainty of the observed DSD mentioned above. A comprehensive and solid validation of the Doppler spectrum simulator require simultaneous and well- aligned DSD and Doppler spectrum measurement, large number of the measured droplet size categories and carefully estimation of the measurement; large number of the measured droplet size categories and carefully estimation of the environment turbulence broadening factors.

**Technical issues**
I have some suggestions for technical corrections, but I am not a native speaker.

Response: we appreciate the reviewer's edits. All the suggested corrections have been made in the revised manuscript.

L22. consistent with

L27. applications for cloud/precipitation

L28. microphysical and dynamical

L34. For a vertical

L35. Provide

Either using positive or negative to indicate downward is fine, but it is appreciated to make a statement in each figure's caption.

Response: we appreciate the reviewer's suggestions. Changes have been made in the caption of Figure 5 and figure 6.

L53. and many other places. Spectral broadening is contributed by a list of factors such as turbulence, horizontal wind, spectral window etc. In some cases, turbulence dominates this broadening effect.

Response: We want to thank the reviewer's comments. This sentence (and may other places) has been modified in the revised manuscript:

Line 53: "…More specifically, the Doppler spectrum width is mainly contributed by the spread of the still-air hydrometeor terminal velocity, the horizontal and vertical wind shear within the radar observation volume, and the environment turbulence…"

Line 390: "…Since the wind shear broadening contribution ($\sigma_S$) to radar Doppler spectrum is generally smaller than $\sigma_D$ and the turbulence broadening ($\sigma_t$) (Borque, Luke et al. 2016), here we neglect the $\sigma_S$ contribution and estimate $\sigma_t$ as…"

L162. This work is published on a journal with which not many cloud radar people familiar, please detail this method.

Response: In the revised manuscript we briefly introduced the turbulent wind generation method and cited the codes we used in this study.

Line 172: "…In this study we adapt the approach proposed by Deodatis (1996) by using the Spectral Representation Method (SRM) to generate the turbulent wind field based on a predefined

Von Karman energy spectrum. The SRM is widely used in the wind engineering community due to its high accuracy, simplicity, and computational efficiency. (Shinozuka and Deodatis 1991, Zhao, Huang et al. 2021). Here, the 1-D turbulence wind is generated with 2 Hz sampling frequency, 1000s duration and with standard deviation of 0.3 ms$^{-1}$, the codes being applied to generate the wind can be accessed from Cheynet (2020)…"

L164-166. This sentence is confusing. Spectrum width is affected by hydrometeor size distribution, how can it be a constant value?

Response: Here we did a theoretical estimation of the turbulence broadening term ($\sigma_t$) by assuming the Doppler spectrum width is only broadened by turbulence. We have added the equation being applied to estimate $\sigma_t$. The sentence has been modified in the revised manuscript:

Line 178: "…The selection of 0.3 ms$^{-1}$ standard deviation is based on a quantitatively estimation of cloud radar observation under a typical cloudy environment. Specifically, for the convective cloud system with eddy dissipation rate ($\varepsilon$) of $5 \times 10^{-3}$ m$^2$ s$^{-3}$ (Mages, Kollias et al. 2022), the turbulence-contributed Doppler spectrum width ($\sigma_t$) from a vertical pointing radar with 30m range resolution($\Delta R$) and 0.3° beamwidth ($\theta$) at 1km height is estimated to be 0.27 ms$^{-1}$ based on the equation from Borque, Luke et al. (2016):

$$\varepsilon \approx \frac{\sigma_t^3}{\sigma_z (1.35\alpha)^{3/2}} \left( \frac{11}{15} + \frac{4}{15} z^2 \frac{\sigma_x^2}{\sigma_z^2} \right)^{-3/2} \tag{1}$$

Where $\alpha$ is the Kolmogorov constant with 0.5, $\sigma_z = 0.35 * \Delta R$, $\sigma_x = \frac{\theta}{4\sqrt{ln2}}$ , $\theta$ is the one-way half-power width with unit of radian. $z$ is height above surface.
…"

L188. close.

Eq.13. where is n in St?

Response: n represents each simulation step.

L292. L306, and many other places. Turbulent environment

L306. echo?

L307. of

L338. spectra from two approaches are consistent with each other.

L340. add a comma before but

L343. approaches

L344. The black line

Figure4. dB(10log10 (mm6 m-3))

Response: Changes have been made in the updated figure.

Figure4 and many other places. Simulated approach looks strange to me. I would call it physics-based approach.

Response: We have renamed the proposed method as Physics-Based Simulation (PBS) approach.

Line 338: "…the broadening of the right edge of the radar Doppler spectrum from the physics-based simulation (PBS) approach…"

L4225. Can be employed in more studies

**Reference**

Borque, P., et al. (2016). "On the unified estimation of turbulence eddy dissipation rate using Doppler cloud radars and lidars." Journal of Geophysical Research: Atmospheres **121**(10): 5972-5989.

Cheynet, E. (2020). Wind field simulation (text-based input), Zenodo, Tech. Rep., 2020, doi: 10.5281/ZENODO. 3774136.

Deodatis, G. (1996). "Simulation of ergodic multivariate stochastic processes." Journal of engineering mechanics **122**(8): 778-787.

Mages, Z., et al. (2022). "Surface-based observations of cold-air outbreak clouds during the COMBLE field campaign." Atmospheric Chemistry and Physics Discussions: 1-39.

Shinozuka, M. and G. Deodatis (1991). "Simulation of stochastic processes by spectral representation."

Wang, D., et al. (2021). Analysis of Three Types of Collocated Disdrometer Measurements at the ARM Southern Great Plains Observatory, Oak Ridge National Lab.(ORNL), Oak Ridge, TN (United States). Atmospheric ….

Zhao, N., et al. (2021). "Simulation of ergodic multivariate stochastic processes: An enhanced spectral representation method." Mechanical Systems and Signal Processing **161**: 107949.

---

## Author Response (AR2)

**Reviewer 1**

The authors have appropriately addressed all of my concerns and comments. However, a few issues, mostly technical and stylistic, are still present in the manuscript. I recommend their work for publication in AMT after these further minor points are solved. Please find the remaining issues listed below, all line numbers refer to the revised manuscript:

Response: We appreciate the reviewer's detailed edits. All the corrects have been made in the revised manuscript.

- line 39: "Doppler spectrum can be used to simulate rain Droplet Size Distribution", I believe the term "simulate" is used improperly and should be replaced.
Done

- line 51: "fully understanding" → "full understanding"
Done

- line 196: the text states that the simulation is set up "with initial velocity set as 0 ms-1"; however the velocities displayed in Fig. 4 are not equal to 0 ms-1 at time = 0s. This needs to be clarified/fixed.

Response: We thank the reviewer for the detailed observation. Figure 4a shows the simulated duration from 10 to 110s. We want to apologize for the oversight.

- line 223, within the caption of Fig. 4: "velocity filed" → "velocity field"
Done

- line 249: I believe "Lhermitte, 2002" should not be in parentheses.
Done

- line 253: "m2 m-3 (ms -1 )" → "m2 m-3 (ms -1 )-1"
Done

- lines 259-262: following the revision the concepts illustrated at these lines are now explained also at lines 178-187. It's not necessary to repeat them.

Response: We have rephrased the sentence as follows:
"…The contribution of turbulence on Doppler spectrum broadening is commonly parameterized as $\sigma_t$…"

- line 277: "specifc" → "specific"
Done

- line 293: "depended" → "dependent"
Done

- line 296: "configerations to guide the chosen" → "configurations to guide the choice"
Done

- line 298: "appromiately" → "approximately"
Done

- lines 311-313: I find this sentence unclear and I recommend it is rephrased.

Response: This sentence has been rephrased as follows:

Specifically, according to the Mie scattering theory, radar backscattering cross section varies in an oscillatory manner with particle size (Mie, 1908). With the 3.2 mm wavelength radar, the backscattering cross section as a function of droplet size is characterized as several local minimal values with diameter of 1.66, 2.86 mm, which are corresponding to still-air terminal fall velocity of 5.83, 7.89 ms-1. This unique feather is known as "Mie notches" in the radar Doppler spectrum (Kollias et al., 2002;Kollias et al., 2007;Courtier et al., 2022).

- line 357: "both two simulators" → "both simulators"
Done

- Figure 6: I believe it is not explained what the blue area represents. I assume it represents the uncertainty in the Doppler spectrum produced by the uncertainty in sigma, but it should be anyhow explained in the caption.

Response: We have added the description in the caption.
"…The blue shaded region represents the uncertainty of the simulated Doppler spectrum produced by the uncertainty in $\sigma_t$ based on the convolution method…."

- line 461: the term "resolving" is used inappropriately here, and I recommend it is replaced with "simulating".
Done

- lines 473-474: "simulator can correct ..." and "simulator can also be utilized …", since the simulator performance has not been validated I recommend that the authors refrain from such certain statements and phrase these sentences more carefully.

Response: We have deleted these two sentences in the revised manuscript.

- line 535, within the Cheynet (2020) reference: in the doi there is a space after the dot.

Response: Corrections have been made in the revised manuscript.

**Reviewer 3**

**The current version is easier to follow. But there are still questions to address.**

1. Fig. 1. A fit was made to the data given in Gunn and Kinzer (1949) which is known for a milestone of raindrop velocity measurement. But in their work, C_d and Re were calculated instead of directly measured. Loth (2008) has made a review on this topic. Have you checked the applicability of C_d and Re of Gunn and Kinzer (1949)?

Response: We thank the reviewer for providing the review paper, which is helpful for consolidating our work. The review from Loth (2008) deals with more complicated shapes of solid particles instead of raindrop which can be considered as perfect sphere when the diameter is small and oblate particles beyond. Instead, the experiment-based estimation of Cd and Re from the Gunn and Kinzer (1949) is widely applied in the cloud physics and radar community, especially for estimating terminal fall velocity from particles (Bartholomew, 2020;Lhermitte, 2002). In Lhermitte (2002, P 91), a comparison is made showing that the Cd estimated from Gunn and Kinzer (1949) is consistent with the one derived from the stokes law as droplet diameter small than 2mm but deviate for large diameter caused by the oblate particle effect. We hereafter considered the experiment-based Cd-Re relationship is applicable for our objective. Moreover, the diameter-terminal fall velocity relationship we adapted in our simulator (i.e. Eq. 9) is also derived from the Gunn and Kinzer (1949), the adapted fitted Cd-Re makes the dynamical framework of the simulator self-consistence. A detailed validation of the Cd-Re relationship can also be found at the interactive discussion with one of the reviewers:
https://editor.copernicus.org/index.php?_mdl=msover_md&_jrl=400&_lcm=oc108lcm109w&_a cm=get_comm_sup_file&_ms=109202&c=242211&salt=14626878041751600723

2. Fig.5. It is good that different turbulence intensities were assumed. In (a), the turbulence is rather weak. We can see that the spectrum generated by the convolution method almost overlaps the still-air condition, as expected. But detectable deviation found in PBS. To my understanding, the inertia effect makes the spectrum of large drops less broadening, therefore better matching with the still-air condition. Why the reddish curves in Fig.5 systematically move towards the slow-falling parts?

Response: We appreciate the reviewer for the detailed observation. However, we would not interpret the PBS-based Doppler spectrum is systematically moving towards the slow-falling part. First, for the right edges, the red line (i.e. PBS-based spectrum) overlaps with the black line instead of the blue line (i.e. convolution-based spectrum), this indicates that even with small $\sigma_t$ as 0.05ms$^{-1}$, the over broadening effect of the Doppler spectrum from the convolution method can be identified. Second, the location of the first notch of the PBS-based spectrum also consistent with other two.

We however noticed the inconsistency between the spectra at the second notch and the left edges. This is due to the generated PBS-based Doppler spectrum is depended on the applied turbulent velocity field, which is generated numerically and may include truncation error regarding the $\sigma_t$ intensity. Nevertheless, the difference of the left edge location between the PBS-

based and the convolution-based spectrum in Fig.5 (a) is around 0.02 ms$^{-1}$, which is considered as the uncertainty of the PBS-based Doppler spectrum. In contrast, what we want to highlight in this study is the over broadening effect which is manifested as the large velocity difference at the right edge which can reach to 1m/s as shown in Fig.5 d.

3. I understand that the authors want to highlight the importance of inertia effect to spectrum simulation. However, the conclusions left me the impression that if the inertia effect is considered, we are good with the spectrum simulation. As I commented before, the current study assumes the absence of many other factors contributing to the spectrum broadening. If all factors are considered, would not the convolution method work? As far as I could see, Tridon et al have a serial of works using the convolution method, and they obtained rather good results. I believe the adequacy of the convolution method should be discussed.

**Response:**

We thank the reviewer's comments. We have modified the conclusions in the revised manuscript. Specifically, we agree with the reviewer that the convolution approach is a well-established and an efficient method to simulate Doppler spectrum, particular in the light precipitation and weak turbulence condition:
"…In the case of light precipitation mostly happening in marine boundary layer cloud, the droplet inertia effect on Doppler spectrum is negligible and the traditional simulator generates consistent results with the proposed simulator…"

The objective of our work is to bring awareness to the community that in the presence of heavy precipitation and especially strong turbulence environment, the over broadening issue of the convolution approach should be considered.

"…The comparison indicates that the traditional Doppler simulator without considering the inertial effect generates an artificially broader Doppler spectrum. This inertia effect becomes more noticeable as turbulence intensity increases. This finding suggests that special caution should be taken when applying convolution-based approaches to represent DSD-turbulence interaction in heavy precipitation…"

Overall, we think this work may be helpful for the radar and cloud physics community for a better interpretation of the results generated from the Doppler spectrum simulator. We have also published our simulator codes at https://doi.org/10.5281/zenodo.7897981 and anticipated more upcoming applications.

**Reference**

Bartholomew, M.: Laser Disdrometer Instrument Handbook, DOE Office of Science Atmospheric Radiation Measurement (ARM) Program …, 2020.
Lhermitte, R. M.: Centimeter & millimeter wavelength radars in meteorology, Lhermitte Publications, 2002.